# Stochastic Dynamic Analysis of a Three-Tailed Helical Microrobot in Confined Spaces

**DOI:** 10.3390/mi16040373

**Published:** 2025-03-25

**Authors:** Xinpeng Shi, Yongge Li, Kheder Suleiman

**Affiliations:** 1School of Mathematics and Statistics, Northwestern Polytectnical University, Xi’an 710072, China; xinpengshi16@163.com (X.S.); liyonge@nwpu.edu.cn (Y.L.); 2Research and Development Institute of Northwestern Polytectnical University in Shenzhen, Shenzhen 518063, China

**Keywords:** dynamic behavior, three-tailed helical microrobots, stochastic dynamic model

## Abstract

This study investigates the complex dynamic behavior of three-tailed helical microrobots operating in confined spaces. A stochastic dynamic model has been developed to analyze the effects of input angular velocity, current, fluid viscosity, and channel width on their motion trajectories, velocity, mean squared displacement (MSD), and wobbling rate. The results indicate that Gaussian white noise exerts a dispersive driving effect on the motion characteristics of the microrobots, leading to a 49% reduction in their velocity compared to deterministic conditions. Additionally, the time required for microrobots to traverse from the initial position to the bifurcation point decreases by 65% when the current is increased and by 39% when the fluid viscosity is reduced. These findings underscore the importance of optimizing control parameters to effectively mitigate noise impacts, enhancing the practical performance of the microrobots in real-world applications. This research offers solid theoretical support and guidance for the deployment of microrobots in complex environments.

## 1. Introduction

As a new microintelligent system, microrobots demonstrate significant potential for applications in biomedicine [1], environmental monitoring [2], micro–nano manufacturing [3], and other fields. They are capable of performing complex tasks in microscale environments, such as targeted drug delivery [4], cell manipulation [5], and microchannel cleaning [6], thereby providing innovative solutions for the advancement of modern science and technology. However, microrobots encounter numerous challenges in practical applications, including motion control, environmental adaptability, and stability [7,8,9]. Therefore, it is essential to conduct in-depth studies on the motion mechanisms and behavioral characteristics of microrobots in complex environments to enhance their performance and effectiveness across various applications.

The research on three-tailed spiral microrobots introduces a novel design concept in the field of microrobotics, elucidating the critical influence of tail structure on motion behavior and establishing a robust theoretical foundation for the optimal design of microrobots. Single-tailed microrobots primarily generate unidirectional propulsion through the rotation of a tail screw, enabling linear motion [10,11]. While this simple structure is easy to design and control, its adaptability and maneuverability in complex environments are limited, making it difficult to meet practical application requirements. Dual-tailed microrobots, on the other hand, achieve steering motion in a two-dimensional plane through the coordinated action of two tails, significantly enhancing motion flexibility and control accuracy [12,13]. However, the dual-tailed structure still faces limitations in three-dimensional motion, struggling to cope with complex three-dimensional fluid environments or spatial tasks. In contrast, the three-tailed spiral design not only significantly improves motion stability, controllability, and adaptability but also enables spiral motion and complex trajectory control in three-dimensional space through the cooperative rotation and asymmetric driving of the three tails. This three-dimensional motion capability expands the application scenarios of microrobots in targeted therapy and micromedical equipment operation, providing new possibilities for efficient navigation and task execution in complex environments.

In practical applications, microrobots usually need to perform tasks in limited environments, such as micropipes, blood vessels, or complex fluid networks [14,15,16]. These confined spaces have a significant impact on the motion behavior of the robots. First of all, the boundary effect will limit the freedom of motion of the robot, making it difficult to extend freely or change direction flexibly, thus affecting its trajectory and overall efficiency [17,18]. Secondly, the geometric shape of the channel (such as bending, bifurcation, or narrow areas) may further change the motion mode of the microrobot and even cause it to collide with the boundary, increasing the instability of the motion [19,20]. In addition, fluid resistance is particularly prominent in restricted environments, especially in high-viscosity fluids, which will significantly slow down the robot’s movement speed and make it difficult to complete tasks efficiently [21,22]. Studying the movement behavior of microrobots in restricted environments, can not only help to deeply uncover the interaction mechanism between microrobots and the environment but also provide a critical theoretical foundation for optimizing the structural design and control strategy of robots, thereby significantly improving their adaptability and performance in complex tasks.

When studying microrobots, it is very important to consider the influence of thermal noise to accurately describe their dynamic characteristics. Due to the microscale and complex working environments of microrobots, such as fluid turbulence, temperature fluctuations, and interactions with surrounding obstacles, they are easily disturbed by various random disturbances [23,24,25,26]. These disturbances may not only lead to deviations in the robot’s trajectory but also cause instability in the system, significantly increasing the error in control and observation. Noise significantly impacts the speed and directional control of microrobots, making it challenging to maintain the desired trajectory in practical applications [27]. Random noise also disturbs the sensing signals of microrobots, reducing their environmental perception and consequently affecting their navigation accuracy in complex environments [28]. Furthermore, the influence of noise on the cooperative movement behavior of microrobots is analyzed, providing a theoretical foundation for developing more robust group control algorithms [29]. These studies not only provide technical support for the application of microrobots in biomedicine, environmental monitoring, and other fields but also lay a solid foundation for their efficient navigation and task execution in complex environments.

In this study, we investigated a microrobot equipped with communication magnetic spheres and three rigid tails, aiming to explore its dynamic behavior under Gaussian white noise excitation. By developing a stochastic dynamic model, we analyzed the effects of input angular velocity, current, fluid viscosity, and channel width on the motion trajectories and velocities of the microrobot in confined spaces in order to evaluate the system’s performance.

The organization of this paper is as follows: The stochastic dynamic model of the microrobot is presented in Section 2. The motion behavior of the microrobot in random environments is analyzed in Section 3. Closing remarks are offered in Section 4.

## 2. Model

In this study, we analyzed the motion characteristics of a three-tailed spiral microrobot operating in a confined space, as illustrated in Figure 1a. Under low Reynolds number conditions, the inertial effect can usually be ignored because fluid resistance dominates the motion. However, to ensure the integrity and accuracy of the theoretical model, we introduce the inertial effect into the microrobot model. Although the inertial effect is small at the microscale, its inclusion allows for a more comprehensive description of the robot’s motion behavior.

The microrobot consists of a communication magnetic sphere and three rigid helical tails of equal length. Figure 1b,c depict the geometric configuration and force analysis. The communication magnetic sphere is driven by an external magnetic field to generate propulsion and torque, while the three rigid helical tails interact with the fluid through rotational motion, further enhancing the propulsion efficiency and stability of the microrobot. In the model, we also consider the influence of confined spaces and thermal noise environments. Confined spaces impose geometric constraints and fluid resistance effects on the motion of the microrobot, while thermal noise introduces random disturbances, affecting the trajectory and stability of the microrobot. The translational and rotational equations of the microrobot can be described as follows [10]:(1)$$\begin{array}{cc} \; & m\ddot{x}=F_{M}+F_{t_{1}}+F_{t_{2}}+F_{t_{3}}+F_{c}-F_{px}+\xi_{1}(t),\\ \; & J\ddot{\theta }=T_{M}+T_{t_{1}}+T_{t_{2}}+T_{t_{3}}+T_{c\theta }-T_{p\theta }+\xi_{2}(t),\\ \; & J\ddot{\Psi }=\sqrt{3}/2\left(F_{t_{3}}-F_{t_{2}}\right)t_{b}+T_{c\Psi }-T_{p\Psi }+\xi_{3}(t),\\ \; & J\ddot{\varphi }=F_{t_{1}}t_{b}-\left(F_{t_{2}}+F_{t_{3}}\right)t_{b}/2+T_{c\varphi }-T_{p\varphi }+\xi_{4}(t), \end{array}$$
where *x* represents the forward displacement, θ denotes the rotation angle of the x axis, Ψ is the rotation angle of the y axis, and φ is the rotation angle of the z axis. $F_{M}$ is the magnetic force, $F_{t_{i}}$, i=1,2,3, represents the force of the tail, and $F_{px}$ denotes the resistive force of the magnetic ball. $t_{b}$ is the offset distance. $T_{M}$ is the magnetic torque; $T_{p\theta }$, $T_{p\Psi }$, and $T_{p\varphi }$ denote the resistive torques from the sphere on the x axis, y axis, and z axis; and $T_{t_{i}}$, i=1,2,3 denotes the torque of the tail. $\xi_{1}(t)$, $\xi_{2}(t)$, $\xi_{3}(t)$, and $\xi_{4}(t)$ are random force and random torque.

*m* represents the sum of the masses of a magnetic sphere $m_{p}$ and the tail $m_{t}$, where $m_{p}=4\pi r^{3}\rho_{p}/3,m_{t}=3\pi b^{2}L_{t}\rho_{t},$ and $\rho_{t}$ and $\rho_{p}$ are the densities of $m_{t}$ and $m_{p}$. The moment of inertia $J=2m_{p}r^{2}/5+3m_{t}b^{2}/2$. *r* is the radius of the magnetic sphere, $L_{t}$ is the length of the tail, and *b* denotes the cross-sectional radius of the tail.

The motion of the microrobot is driven by an external rotating magnetic field, which applies torque to the magnetic components of the structure, including the spherical body and the helical tail. The external magnetic field is generated by a set of electromagnetic coils and rotates in a controlled manner. As the magnetic field rotates, it induces the magnetic moments of the microrobot to align with the field direction, resulting in a torque that causes both the spherical body and the helical tail to rotate [30]. The magnetic field is generated and controlled using a Helmholtz coil system. The zero point of the coordinate system is set at the intersection of the coil axis and the central plane of the Helmholtz coils. Due to the influence of the uniform magnetic field, the magnetic force is zero, i.e., $F_{M}=0$. The resulting magnetic torques are given by the following equations:(2)$$\begin{array}{cc} \; & T_{m}=V_{p}MB, \end{array}$$
where $V_{P}$ is the volume of the communication magnetic sphere, *M* is the magnetization, the magnetic field strength $B=\mu_{0}R^{2}nI/2\{1/{\left[R^{2}+{\left(x+R/2\right)}^{2}\right]}^{3/2}+1/{\left[R^{2}+{\left(x-R/2\right)}^{2}\right]}^{3/2}\}$, *R* is the mean radius of the coil, *n* is the coil turns, $\mu_{0}$ is the permeability of free space, and *I* is the current.

The tails generate the force and torque, which can be expressed as(3)$$\begin{array}{cc} \; & F_{t_{i}}=-L_{t}(C_{tn}\sin^{2}\beta +C_{tt}\cos^{2}\beta )\sec\beta \dot{x}+L_{t}(C_{tt}-C_{tn})A_{t}\sin\beta (\omega_{i}-\dot{\theta }),\\ \; & T_{t_{i}}=M_{t_{i}}+M_{t_{i}v},i=1,2,3, \end{array}$$
where $\omega_{i},i=1,2,3$ is the input angular velocity of the microrobot system and $\dot{\theta }$ is the resistive angular velocity. The driving torque $M_{t_{i}}=L_{t}(C_{tt}-C_{tn})A_{t}\sin\beta \dot{x}+L_{t}(C_{tt}\sin^{2}\beta +C_{tn}\cos^{2}\beta )A_{t}^{2}\sec\beta (\omega_{i}-\dot{\theta }),i=1,2,3$. The resistive coefficients $C_{tt}=2\pi \mu /[\ln(2\lambda_{t}/b)-1/2]$, $C_{tn}=4\pi \mu /[\ln(2\lambda_{t}/b)+1/2]$. $A_{t}$ is amplitude of the tail, $\beta =\arctan(2\pi A_{t}/\lambda_{t})$ is pitch angle of the tail, $\lambda_{t}$ represents wavelength of the tail, μ is the fluid viscosity, and $M_{t_{i}v}=4\pi \mu b^{2}L_{t}\cos\beta (\omega_{i}-\dot{\theta }),i=1,2,3$ is the viscous torque.

The resistive force and resistive torque induced by the communication magnetic sphere can be written in the following form:(4)$$\begin{array}{cc} \; & F_{px}=6\pi \mu r\dot{x},\\ \; & T_{p\theta }=8\pi \mu r^{3}\dot{\theta },\\ \; & T_{p\Psi }=8\pi \mu r^{3}\dot{\Psi },\\ \; & T_{p\varphi }=8\pi \mu r^{3}\dot{\varphi }. \end{array}$$

In the process of a microrobot making contact with a wall in a confined space, the contact force and contact torque are critical physical quantities that describe the interaction between the two. The contact process can be divided into two stages, loading and unloading, which correspond to the microrobot gradually approaching and separating from the wall [31].

The loading process refers to the phase in which the contact force gradually increases as the microrobot approaches the wall and begins to make contact. The contact force can be expressed as(5)$$\begin{array}{cc} \; & F_{c}=-k{\left|\delta_{1}\right|}^{p}H(\delta_{1}), \end{array}$$
where $k=\frac{\frac{4\sqrt{r}}{3}}{\frac{1\;-\;\sigma_{c}^{2}}{E_{c}}+\frac{1\;-\;\sigma_{w}^{2}}{E_{w}}}$ represents the contact stiffness, which is calculated based on the Hertz contact theory. $E_{c}$ and $E_{w}$ represent the Young’s modulus of the microrobot and the blood vessel wall, respectively, while $\sigma_{c}$ and $\sigma_{w}$ denote their respective Poisson’s ratios. $\delta_{1}$ represents the elastic deformation of the wall at the contact point, while $H(\delta_{1})$ denotes the Hayside step function, which is utilized to describe the effective conditions of the contact force. During the loading process, the elastic deformation $\delta_{1}$ increases with time, indicating that $\dot{\delta_{1}}>0$.

The unloading process refers to the gradual separation of microrobots from a surface after contact, during which the contact force decreases until complete detachment occurs. The contact force during unloading can be expressed as follows:(6)$$\begin{array}{cc} \; & F_{c}=-F_{\delta_{m}}{\left|\frac{\delta_{1}-\delta_{0}}{\delta_{m}-\delta_{0}}\right|}^{p}H(\delta_{1}), \end{array}$$
where $F_{\delta_{m}}$ is the maximum contact force, $\delta_{m}$ denotes the corresponding maximum wall deformation, and $\delta_{0}$ indicates the permanent deformation resulting from loading or unloading. The index p∈{1.5;2.5}. During the unloading process, the elastic deformation $\delta_{1}$ decreases over time, indicating that $\dot{\delta_{1}}<0$.

Contact torque can be calculated using the cross product of the contact force and the position vector of the contact point:(7)$$\begin{array}{cc} \; & T_{c}=F_{c}\times r_{c}. \end{array}$$

We propose that the positional change of the contact point is primarily due to elastic deformation such that the position vector $r_{c}$ approximates the elastic deformation $\delta_{1}$.

Assuming the microrobot is influenced by thermal noises $\xi_{1}(t)$, $\xi_{2}(t)$, $\xi_{3}(t)$, and $\xi_{4}(t)$ [32], these noises are mutually independent and can be represented as Gaussian white noise with a mean of zero and intensities $D_{i},i=1,2,3,4$,(8)$$\begin{array}{cc} \; & \langle \xi_{i}(t)\rangle =0,\;\langle \xi_{i}(t)\xi_{i}(t_{0})\rangle =2D_{i}\delta (t-t_{0}),i=1,2,3,4, \end{array}$$
where δ(·) means the Dirac-delta function and $\langle \cdot \rangle$ is the expected value.

Replacing Equations (Equation 2)–(Equation 4) in Equation (Equation 1), the system models of the microrobot can be represented in the following manner:(9)$$\begin{array}{cc} \; & m\ddot{x}=A_{11}\dot{x}+A_{12}(\omega_{1}+\omega_{2}+\omega_{3}-3\dot{\theta })+F_{c}+\xi_{1}(t),\\ \; & J\ddot{\theta }=T_{M}+3A_{12}\dot{x}+A_{13}(\omega_{1}+\omega_{2}+\omega_{3}-3\dot{\theta })-8\pi \mu r^{3}\dot{\theta }+T_{c\theta }+\xi_{2}(t),\\ \; & J\ddot{\Psi }=\sqrt{3}A_{12}t_{b}(\omega_{3}-\omega_{2})/2-8\pi \mu r^{3}\dot{\Psi }+T_{c\Psi }+\xi_{3}(t),\\ \; & J\ddot{\varphi }=A_{12}t_{b}(\omega_{1}-\dot{\theta })-t_{b}A_{12}(\omega_{2}+\omega_{3}-2\dot{\theta })/2-8\pi \mu r^{3}\dot{\varphi }+T_{c\varphi }+\xi_{4}(t), \end{array}$$
where(10)$$\begin{array}{cc} A_{11} & =-[3L_{t}(C_{tn}\sin^{2}\beta +C_{tt}\cos^{2}\beta )\sec\beta +6\pi \mu r],\\ A_{12} & =L_{t}(C_{tt}-C_{tn})A_{t}\sin\beta ,\\ A_{13} & =L_{t}(C_{tt}\sin^{2}\beta +C_{tn}\cos^{2}\beta )A_{t}^{2}\sec\beta +4\pi \mu b^{2}L_{t}\cos\beta . \end{array}$$

The motion equations presented above are formulated in the object coordinate system. To provide a more intuitive representation of the microrobot’s motion, these equations are transformed into the reference coordinate system X-Y-Z, as illustrated in Figure 2. In this reference frame, the forward displacement x is decomposed into three components:(11)$$\begin{array}{cc} X & =x\cos\Psi \cos\varphi ,\\ Y & =x\cos\Psi \sin\varphi ,\\ Z & =-x\sin\Psi . \end{array}$$

## 3. Simulation Results

In this study, we employ the fourth-order stochastic Runge–Kutta algorithm in MATLAB (version R2023a) to numerically simulate the trajectories of microrobots under varying input angular velocities. The effects of input angular velocity, current, fluid viscosity, and channel width on the speed, mean square displacement (MSD), and swing rate of the microrobots are systematically investigated. The regulatory mechanisms of these parameters on the motion behavior of the microrobots are revealed. Furthermore, a detailed analysis of the trajectories of microrobots in bifurcated pipelines is conducted, examining the time characteristics from their initial positions to the bifurcation points. The geometric parameters of the microrobots are provided in Table 1.

### 3.1. The Influence on the Trajectory

By controlling the input angular velocities of the microrobot’s tails, the trajectory variations in different spatial environments can be analyzed, thereby revealing the physical relationship between motion patterns and spatial constraints.

Only the confined space is considered as a cylindrical channel, where the channel radius $r_{m}=0.003\;m$. In Figure 3a, the blue solid line and orange dashed line represent the motion trajectories in free space and confined space, respectively. When the input angular velocities of all tails are identical (i.e., the rotational speeds of the tails remain consistent), the microrobot moves in a straight line along its principal axis. This is because the symmetric angular velocities of the tails generate balanced propulsion forces, preventing any deviation in the overall motion direction and resulting in a stable linear motion. However, in the confined space considered, the microrobot does not collide with the boundaries while moving along the straight path, thus maintaining linear motion under limited constraints.

As shown in Figure 3b, when the input angular velocities of two tails are identical, the microrobot exhibits a helical motion trajectory. However, in confined space, geometric constraints restrict the degrees of freedom of its helical motion, reducing the radius and increasing the twist of the helical trajectory, which manifests as a contraction of the motion range. This demonstrates the significant influence of spatial boundaries on free motion at the microscale, highlighting the boundary effects on the motion characteristics of microrobots in confined environments.

Figure 3c illustrates that when the input angular velocities of all three tails differ, the microrobot demonstrates a more complex and irregular curved motion. This behavior is attributed to the asymmetry in both the direction and magnitude of the thrust produced by the three tails. Although the motion path appears random, the trajectory remains controllable through the regulation of the input angular velocities. By adjusting the angular velocities of the three tails, fine control over the motion trajectory of the microrobot can be achieved. In confined space, the cylindrical boundaries limit the motion amplitude, resulting in a smaller range of irregular motion compared to free space. This configuration not only enhances the motion flexibility of the robot in narrow spaces but also enables controlled motion ranges and trajectories through different combinations of angular velocities.

When Gaussian white noise is introduced to simulate random disturbances in the environment, the motion of the microrobot becomes uncontrollably more irregular, as shown in Figure 3d. This phenomenon reveals the significant impact of random disturbances in microscopic systems, indicating that under noise influence, the motion behavior of the microrobot becomes difficult to predict and may deviate from the intended trajectory. Such deviations can affect the precise control and motion stability of the microrobot in confined environments. Therefore, in-depth analysis of the motion characteristics of microrobots under random environmental conditions is of great importance.

### 3.2. The Influence on the Velocity

By analyzing how different parameters influence velocity variations, the contributions of various forces to motion can be directly observed, thereby enhancing the understanding of the relationship between force and velocity. As demonstrated in Figure 4, an increase in current directly amplifies the magnetic field strength, which in turn increases the magnetic torque acting on the microrobot, enabling it to overcome fluid resistance and achieve higher motion speeds. On the other hand, as fluid viscosity increases, the viscous drag force exerted by the fluid on the microrobot gradually rises, significantly counteracting its forward velocity. The increase in fluid viscosity implies that the microrobot must overcome greater resistance during motion, leading to a gradual reduction in its speed. Furthermore, within a certain range, the width of the channel has no significant impact on the microrobot’s velocity, as the microrobot primarily relies on magnetic propulsion rather than being constrained by channel boundaries. Compared to deterministic driving conditions, the motion speed of the microrobot decreases in a random environment. Random driving introduces irregular fluctuations in force and torque, disrupting the stable motion trajectory of the microrobot. These random disturbances weaken the continuous driving force, causing the average speed of the microrobot to gradually decline.

### 3.3. The Mean Squared Displacement

The influence of different input angular velocities and channel widths on the MSD of microrobots was analyzed to better understand their diffusion behavior in confined environments, highlighting the diffusive characteristics of microrobot motion. As shown in Figure 5a, the MSD of the microrobots significantly increases with the expansion of the channel radius. This is attributed to the larger activity space provided by wider channels, which enhances the displacement range of the microrobots. In Figure 5b, when the input angular velocities are identical, the MSD of the microrobots is relatively small due to the balanced driving forces, resulting in more stable motion trajectories and a reduced diffusion range. When two input angular velocities are identical, the microrobots experience asymmetric driving forces, leading to helical motion and an increase in MSD. When all three input angular velocities are different, the motion of the microrobots becomes highly irregular, and the unbalanced driving forces significantly enhance their diffusivity, causing the MSD to reach its maximum value. The variations in MSD reflect the diffusion characteristics of microrobots under different driving conditions and channel constraints, revealing the influence of driving forces and spatial degrees of freedom on their motion in confined environments. These insights are crucial for predicting and controlling the behavior of microrobots in complex environments.

### 3.4. The Wobbling Rate

By analyzing the influence of parameter variations on the wobbling rate of microrobots, the generation mechanism and regulation of their periodic motion in microscopic environments can be understood. Different tasks and environmental requirements may necessitate adjustments to the wobbling rate to meet specific motion demands. Studying the wobbling rate helps ensure that microrobots can effectively navigate through narrow or irregular spaces. The wobbling rate is defined as $\omega_{S}=\sqrt{\omega_{\Psi }^{2}+\omega_{\varphi }^{2}}$ [33], where $\omega_{\Psi }$, $\omega_{\varphi }$ represent the rotational angular velocity around the Y axis and the Z axis, respectively.

Figure 6 reveals that as the channel radius gradually increases, the wobbling rate of the microrobots also increases. When the channel width reaches a certain value, the wobbling rate stabilizes. The constraints of the channel directly affect the motion patterns of the microrobots. The larger the space, the higher the possibility and amplitude of oscillation. In Figure 6a, when the channel width increases to a certain extent, the microrobots gain sufficient degrees of freedom during motion, and the oscillation amplitude is no longer limited by the channel size. This implies that even if the channel further widens, the wobbling rate of the microrobots will not significantly increase, as their motion approaches that in free space. When the input angular velocities are identical, the wobbling rate of the microrobots is zero. This indicates that under uniform driving, the microrobots exhibit no relative motion and thus cannot generate oscillations. The uniform driving force does not induce dynamic changes within the microrobots, resulting in the absence of oscillation. When two input angular velocities are identical, the wobbling rate of the microrobots increases. This is due to the asymmetric driving force causing the microrobots to oscillate. This change demonstrates that differences in input angular velocities directly influence the motion state of the microrobots, making their dynamic behavior more complex. When all input angular velocities are different, the wobbling rate reaches its maximum. This indicates that unbalanced driving forces enable the microrobots to oscillate with greater amplitude, increasing their motion freedom and flexibility. This also reflects the importance of driving force imbalance on the behavior of microrobots. Figure 6b demonstrates that as the current increases, the magnetic torque generated by the microrobots strengthens, enhancing the driving force acting on them. The increased driving force improves the directionality of motion, allowing the microrobots to maintain more consistent trajectories within a certain range and reducing the likelihood of large deviations, thereby minimizing the possibility of significant oscillations. As the fluid viscosity increases, the wobbling rate of the microrobots also rises in Figure 6c. Higher viscosity means that the microrobots face greater fluid resistance during motion, which slows down their oscillatory movements and leads to larger oscillation amplitudes. Conversely, in lower-viscosity fluids, the directionality of the microrobots’ motion is enhanced. In this case, the fluid resistance on the microrobots decreases, enabling them to move at higher speeds and with stronger propulsion, maintaining smaller oscillation amplitudes and achieving more stable motion, thus avoiding the impact of large oscillations on their trajectories.

In Figure 6d, the wobbling rate of microrobots in a random environment is lower than that under deterministic conditions within a specific range of channel radii. Deterministic conditions refer to a scenario in which the dynamics of microrobots are modeled without accounting for Gaussian white noise, offering an idealized representation of their behavior. In contrast, random conditions incorporate Gaussian white noise to simulate real-world uncertainties. Under random conditions, microrobots are subjected to random disturbances from different directions and magnitudes, which are temporally disordered. Random forces from different directions tend to cancel each other out, reducing the net motion and oscillation amplitude of the microrobots, resulting in an overall decrease in the wobbling rate. When the channel radius increases beyond a certain point, the difference between random and deterministic conditions gradually diminishes. This is because in larger spaces, the interaction and cancellation effects of disturbances become less significant, allowing the microrobots to move more freely, and their wobbling rate approaches that in deterministic conditions. This result highlights the importance of considering environmental randomness in practical applications, where microrobots may face various uncertainties and disturbances. This understanding aids in optimizing the control strategies and performance of microrobots, achieving better stability and efficiency.

### 3.5. Motion Trajectory in Bifurcated Channels

By investigating the motion trajectories of microrobots in bifurcated channels and their behavior under the influence of noise through numerical simulations and statistical analysis, this study reveals the regulatory mechanisms of driving conditions on motion patterns. The channel produces two bifurcations at a distance of 0.0035m from the starting point. The outer boundary of the cylindrical channel is defined by the following function:(12)$$\begin{array}{cc} Z=\pm 3\times 10^{-3},\; & X\leq 0.0035. \end{array}$$

The outer boundary of the bifurcated channel is represented by a linear function:$$Z=\left\{\begin{array}{cc} 3\times 10^{-3}+0.5(X-0.0035), & X>0.0035,\;\mathrm{the}\;\mathrm{upper}\;\mathrm{boundary},\\ -3\times 10^{-3}-0.5(X-0.0035), & X>0.0035,\;\mathrm{the}\;\mathrm{lower}\;\mathrm{boundary}. \end{array}\right.$$

The inner boundary of the bifurcated channel is defined by the following function:(13)$$\begin{array}{c} Z=\pm 0.5(X-0.0035),\;X>0.0035. \end{array}$$

In our study, we conducted a statistical analysis of the movement trajectories of microrobots over identical time intervals. A total of 10,000 distinct data sets were collected to enable a comprehensive investigation into their spatial distribution characteristics. As shown in Figure 7a, when the input angular velocities of all three tails are identical, the microrobot maintains linear motion and enters the upper and lower bifurcating channels with equal probability at the bifurcation point. This indicates that the microrobot can maintain motion stability under symmetric driving conditions. In Figure 7b, when the input angular velocities of two tails are identical and greater than that of the third tail, the microrobot exhibits helical motion, collides with the lower boundary of the channel, and subsequently enters the lower bifurcating channel. Similarly, as illustrated in Figure 7c, when the input angular velocities of two tails are less than that of the third tail, the microrobot collides with the upper boundary and enters the upper bifurcating channel. This behavior demonstrates that asymmetric driving conditions generate a net torque on the microrobot, causing its trajectory to deviate and ultimately guiding it toward a specific direction. According to Figure 7d, when the input angular velocities of all three tails are different, the microrobot exhibits oscillatory motion, illustrating that complex driving conditions may lead to motion instability.

### 3.6. The Time to Reach the Bifurcation

Through numerical simulations, this study systematically investigates the influence of different parameter variations on the time taken for microrobots to reach the bifurcation point, revealing the regulatory mechanisms of driving conditions, fluid properties, and environmental noise on their motion behavior. As shown in Figure 8a, when the input angular velocities of all three tails of the microrobot are identical, the time taken to reach the bifurcation point significantly decreases as the driving current gradually increases. This phenomenon indicates that an increase in driving current enhances the propulsion force of the microrobot, thereby improving its motion speed. However, as the noise intensity increases, the time to reach the bifurcation point is significantly prolonged, demonstrating that environmental noise disrupts the motion stability of the microrobot and reduces its motion efficiency. As shown in Figure 8b, when the fluid viscosity gradually increases, the time taken for the microrobot to reach the bifurcation point also significantly increases. This is due to the greater resistance exerted by high-viscosity fluids on the motion of the microrobot, thereby slowing its speed. Similarly, as the noise intensity increases, the time required to reach the bifurcation point becomes longer.

## 4. Conclusions

In this study, the dynamic behavior of a three-tailed spiral microrobot in a confined space within a random environment is systematically analyzed. The effects of input angular velocity, current, fluid viscosity, and channel width on its trajectory, velocity, MSD, and wobbling rate are discussed. The results show that the microrobot can maintain stable linear motion under symmetric driving conditions. However, its motion trajectory becomes more complex and challenging to predict under the influence of asymmetric driving and random disturbances. In the stochastic case, the velocity of the microrobots decreases to 0.0018 m/s, which is approximately 49% lower than the deterministic case (0.0035 m/s). The research also reveals that the geometric constraints of the channel significantly impact the motion characteristics of microrobots, especially in bifurcated channels. As the driving current increases, the propulsion of the microrobot is enhanced, leading to improved movement speed. Specifically, when the current is increased fivefold, the time taken by the microrobots to travel from the initial position to the bifurcation point is reduced by approximately 65% (from 17 s to 6 s). Similarly, when the fluid viscosity is reduced to one-third, this time is decreased by approximately 39% (from 12.9 s to 7.9 s). Through an in-depth analysis of these factors, this study provides theoretical support for optimizing the control strategies of microrobots in practical applications and lays a foundation for understanding their behavior in complex environments.

## Figures and Tables

**Figure 1 micromachines-16-00373-f001:**
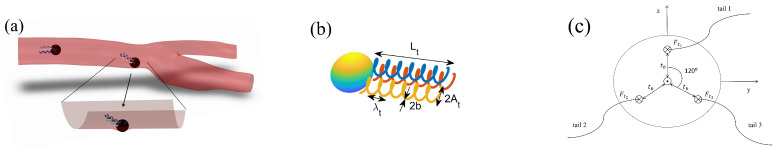
Schematic of the microrobot: (**a**) confined space; (**b**) geometric configuration; (**c**) force analysis.

**Figure 2 micromachines-16-00373-f002:**
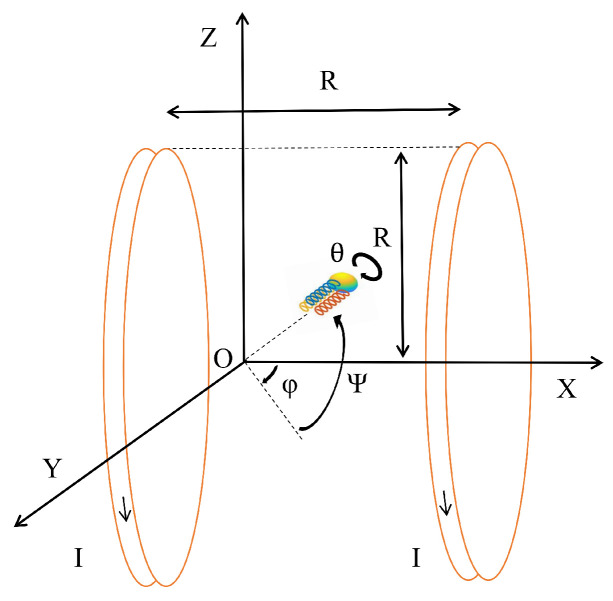
Schematic of the microrobot in reference coordinate system.

**Figure 3 micromachines-16-00373-f003:**
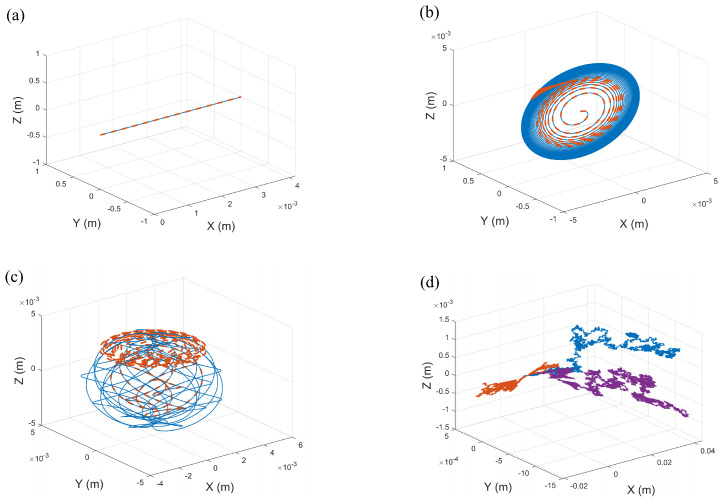
The positional changes of the microrobot: The blue solid line and orange dashed line represent the motion trajectories in free space and confined space, respectively. (**a**) The deterministic case with identical input angular velocities, $\omega_{1}=\omega_{2}=\omega_{3}=5\;\mathrm{rad}/s$. (**b**) The deterministic case where the two tail input angular velocities are the same, $\omega_{1}=5\;\mathrm{rad}/s$, $\omega_{2}=\omega_{3}=40\;\mathrm{rad}/s$. (**c**) The deterministic case in which the three tail input angular velocities differ, $\omega_{1}=5\;\mathrm{rad}/s$, $\omega_{2}=10\;\mathrm{rad}/s$, $\omega_{3}=50\;\mathrm{rad}/s$. (**d**) The stochastic case with identical input angular velocities, $D_{1}=0.00001$, $D_{2}=D_{3}=D_{4}=0.00002$. μ=0.001 Pa·s, I=1 A, $r_{m}=0.003\;m$.

**Figure 4 micromachines-16-00373-f004:**
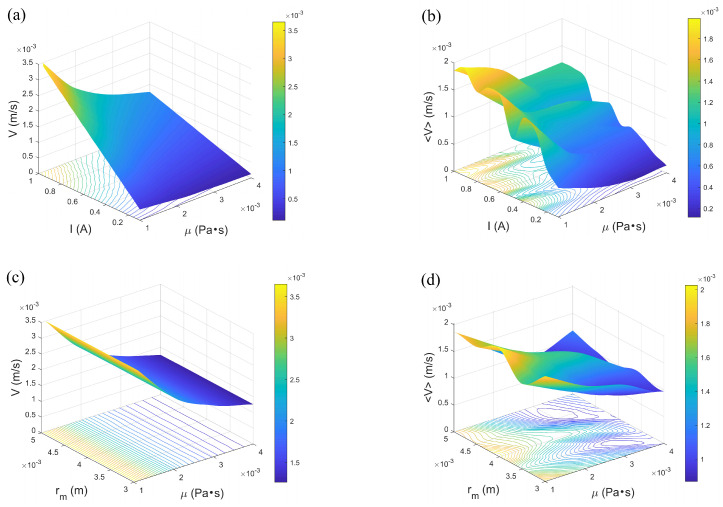
Changes in microrobot speed under the influence of various parameters: (**a**) Variation of current and fluid viscosity under deterministic case, $r_{m}=0.003\;m$. (**b**) Variation of current and fluid viscosity under stochastic case, $r_{m}=0.003\;m$, $D_{1}=0.00001$, $D_{2}=D_{3}=D_{4}=0.00002$. (**c**) Variation in channel width and fluid viscosity under deterministic case, I=1 A. (**d**) Variation in channel width and fluid viscosity under stochastic case, I=1 A, $D_{1}=0.00001$, $D_{2}=D_{3}=D_{4}=0.00002$. $\omega_{1}=5\;\mathrm{rad}/s$, $\omega_{2}=\omega_{3}=40\;\mathrm{rad}/s$.

**Figure 5 micromachines-16-00373-f005:**
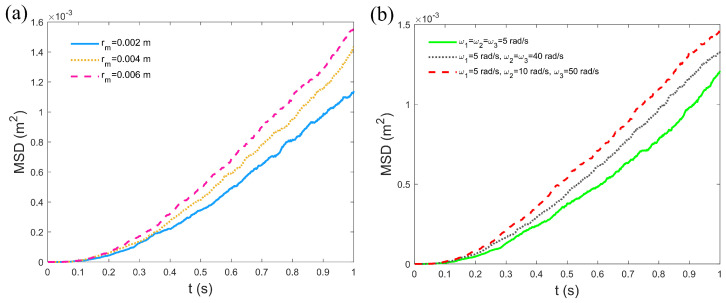
The MSD analysis: (**a**) MSD as a function of time under varying channel widths with $\omega_{1}=5\;\mathrm{rad}/s$, $\omega_{2}=\omega_{3}=40\;\mathrm{rad}/s$. (**b**) MSD as a function of time under varying input angular velocities with $r_{m}=0.004\;m$. Other parameters are set as μ=0.001 Pa·s, I=1 A, $D_{1}=0.00001$, $D_{2}=D_{3}=D_{4}=0.00002$.

**Figure 6 micromachines-16-00373-f006:**
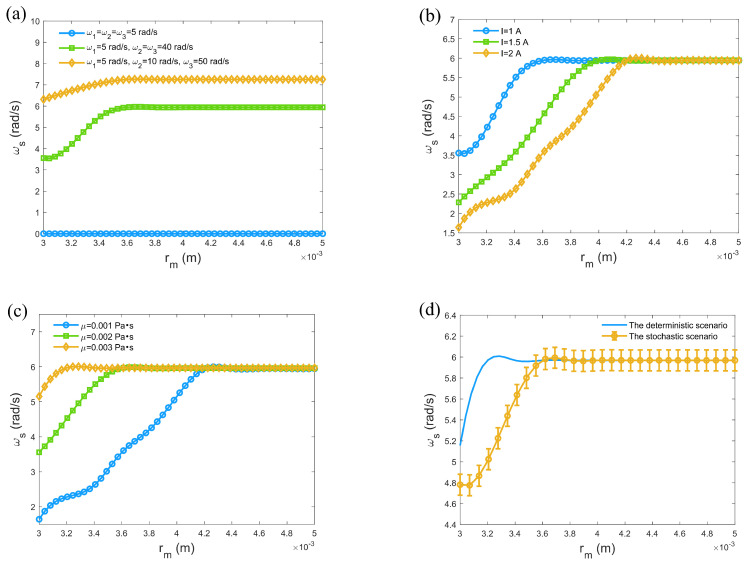
Changes in wobbling rate under the influence of various parameters: (**a**) under varying input angular velocities; (**b**) under varying current; (**c**) under varying fluid viscosity. (**d**) Comparison between stochastic case and deterministic case. μ=0.001 Pa·s, I=1 A, $\omega_{1}=5\;\mathrm{rad}/s$, $\omega_{2}=\omega_{3}=40\;\mathrm{rad}/s$, $D_{1}=0.00001$, $D_{2}=D_{3}=D_{4}=0.00002$.

**Figure 7 micromachines-16-00373-f007:**
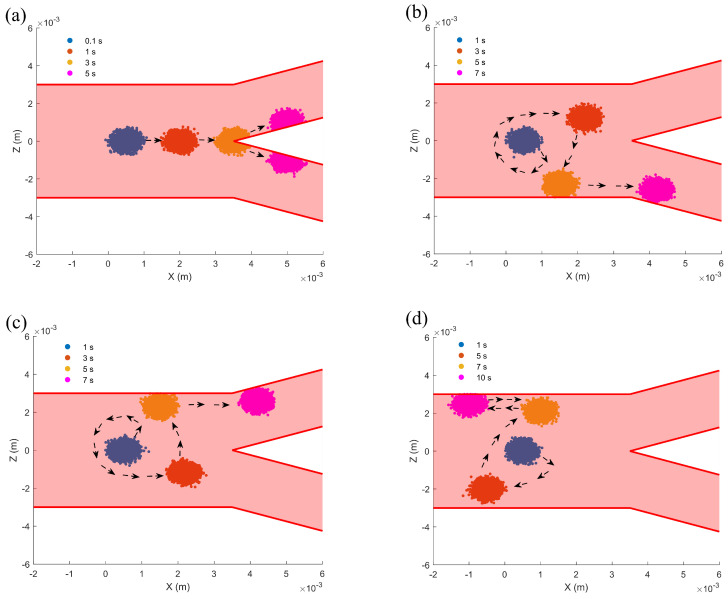
Trajectory analysis of the microrobot in a bifurcated channel: (**a**) Case with identical input angular velocity, $\omega_{1}=\omega_{2}=\omega_{3}=5\;\mathrm{rad}/s$. (**b**) Case with two input angular velocities equal to and greater than the third, $\omega_{1}=5\;\mathrm{rad}/s$, $\omega_{2}=\omega_{3}=40\;\mathrm{rad}/s$. (**c**) Case with two input angular velocities equal to and less than the third, $\omega_{1}=40\;\mathrm{rad}/s$, $\omega_{2}=\omega_{3}=5\;\mathrm{rad}/s$. (**d**) Case where all input angular velocities are different, $\omega_{1}=5\;\mathrm{rad}/s$, $\omega_{2}=10,\omega_{3}=50\;\mathrm{rad}/s$. μ=0.001 Pa·s, I=1 A, $D_{1}=0.00001$, $D_{2}=D_{3}=D_{4}=0.00002$.

**Figure 8 micromachines-16-00373-f008:**
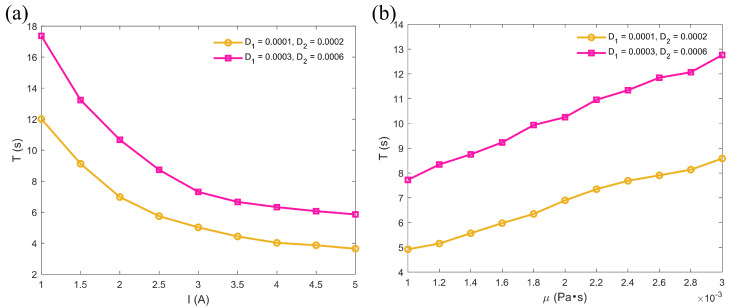
Effect of parameter variations on the microrobot’s time to reach the bifurcation point: (**a**) under different currents, μ=0.001 Pa·s; (**b**) under different fluid viscosities, I=1 A. $\omega_{1}=\omega_{2}=\omega_{3}=5\;\mathrm{rad}/s$.

**Table 1 micromachines-16-00373-t001:** Parameters values.

Parameter Name	Symbol	Value
Radius of the magnetic sphere	*r*	$2\times 10^{-4}\;m$
Length of the tail	$L_{t}$	$8\times 10^{-4}\;m$
Amplitude of the tail	$A_{t}$	$4\times 10^{-4}\;m$
Wavelength of the tail	$\lambda_{t}$	$2\times 10^{-4}\;m$
Maximum wall deformation	$\delta_{m}$	$10^{-4}\;m$
Permanent deformation	$\delta_{0}$	$10^{-6}\;m$
Offset distance	$t_{b}$	$10^{-4}\;m$
Mean radius of the coil	*R*	$8\times 10^{-4}\;m$
Cross-sectional radius of the tail	*b*	$2\times 10^{-5}\;m$
Coil turn	*n*	10N
Poisson’s ratio of the microrobot	$\sigma_{c}$	0.27
Poisson’s ratio of the blood vessel wall	$\sigma_{w}$	0.2
Young’s modulus of the microrobot	$E_{c}$	$10^{9}\;\mathrm{Pa}$
Young’s modulus of the blood vessel wall	$E_{w}$	$10^{5}\;\mathrm{Pa}$
Density of the sphere	$\rho_{p}$	$6500\;\mathrm{kg}/m^{3}$
Density of the tail	$\rho_{t}$	$1500\;\mathrm{kg}/m^{3}$
Permeability of free space	$\mu_{0}$	$4\pi \times 10^{-7}\;T\cdot m/A$
Magnetization	*M*	$2\times 10^{4}\;A/m$

## Data Availability

Data are contained within the article.

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
