# Peer review of "Stochastic Dynamic Analysis of a Three-Tailed Helical Microrobot in Confined Spaces"

_micromachines, 2025, doi:10.3390/mi16040373_

Round 1
Reviewer 1 Report
Comments and Suggestions for Authors
The paper makes significant theoretical contributions to microrobotics through its three-tailed design and stochastic framework. However, its biomedical relevance and practical utility require further experimental and contextual refinement and some comments as follows.
(1)Conduct in vitro experiments to validate the stochastic model.
(2)Integrate non-Newtonian fluid dynamics and energy consumption analysis into the model.
(3)Explore parameter optimization strategies for noise-resistant microrobot designs
(4) Redefine the wobbling rate to include all three tails(ωwobbling = \sqrt(ω12 +ω22 +ω32)​)
(5) The model assumes linear resistive forces but overlooks non-Newtonian fluid effects prevalent in biological systems (e.g., shear-thinning blood). This limits its accuracy for in vivo applications.
(6)While biomedical applications are mentioned, the paper does not explicitly link stochastic modeling to biological phenomena (e.g., Brownian motion in blood plasma, cell-mediated disturbances). A deeper discussion of how noise intensities (Di) relate to physiological conditions would strengthen its biomathematical relevance.
Comments on the Quality of English LanguageThe English could be improved to more clearly express the research.
Reviewer 2 Report
Comments and Suggestions for Authors
I have carefully considered and read the manuscript entitled "Stochastic Dynamic Analysis of a Three-Tailed Helical Microrobot in Confined Spaces" which mainly focuses on investigation of dynamic behavior of three-tailed helical microrobots operating in confined spaces.
The following corrections are recommended.
- In the abstract (lines 8, 9 and 10), you mentioned “Gaussian white noise exerts a dispersive driving effect on the motion of the microrobots”. It should be motion characteristics not only motion. Please correct this sentence.
- In the abstract, results are not properly described. Results should be described in quantitative form and add comparative lines in the abstract.
- In the abstract (line 17), “applications in biomedicine [1],” it seems like bolded words. Please correct this font like others.
- In Introduction section (line 77), you wrote various parameters, what parameters? Please mention the parameters.
- Figure 1 should be placed in the Model section.
- In Figure 1, you presented characteristics of microrobot but how helical tails will be rotated?
- You wrote about external magnetic field but there is no schematic regarding external magnetic field is mentioned and how it will rotate sphere and helical tails.
- Elaborate, on what principle the microrobot will be supported in a confined space.
- In table 1, parameters name should be added. It would be convenient for readers.
- In Simulation results section (lines 166 & 167), what kind of tool do you use for numerical simulations?
- Correct figure 5 captions.
- In Figure 6d, you mentioned about deterministic scenario and stochastic scenario but there is no explanation about these scenarios available in the manuscript. And on what principle do you make these scenarios.
- There is no validation study conducted to validate the present numerical simulation results.
- Sensitivity analysis should be recommended to strengthen the results section.
- The conclusion is lack of necessary findings such as quantitative and comparison information. Improve this section.
Round 2
Reviewer 2 Report
Comments and Suggestions for Authors
The paper may be published.